# PatchTraj: Unified Time-Frequency Representation Learning via Dynamic Patches for Trajectory Prediction

## Abstract

Pedestrian trajectory prediction is crucial for autonomous driving and robotics. While existing point-based and grid-based methods expose two main limitations: insufficiently modeling human motion dynamics, as they fail to balance local motion details with long-range spatiotemporal dependencies, and the time representations lack interaction with their frequency components in jointly modeling trajectory sequences. To address these challenges, we propose PatchTraj, a dynamic patch-based framework that integrates time-frequency joint modeling for trajectory prediction. Specifically, we process trajectories through parallel time and frequency branches, and employ dynamic patch partitioning for multi-scale segmentation, capturing hierarchical motion patterns. Each patch undergoes adaptive embedding with scale-aware feature extraction, followed by hierarchical feature aggregation to model both fine-grained and long-range dependencies. The outputs of the two branches are further enhanced via cross-modal attention, facilitating complementary fusion of temporal and spectral cues. The resulting enhanced embeddings exhibit strong expressive power, enabling accurate predictions even when using a vanilla Transformer architecture. Extensive experiments on ETH-UCY, SDD, NBA, and JRDB datasets demonstrate that our method achieves state-of-the-art performance. Notably, on the egocentric JRDB dataset, PatchTraj attains significant relative improvements of 26.7% in ADE and 17.4% in FDE, underscoring its substantial potential in embodied intelligence.

## 1 Introduction

Pedestrian trajectory prediction is critical for autonomous driving (Jiang et al., 2023; Chai et al., 2019; Park et al., 2024) and robotics (Karnan et al., 2022; Zhu et al., 2021), where accurately forecasting future paths from observed trajectories hinges on effectively modeling spatiotemporal dependencies. Existing methods predominantly perform temporal-domain modeling of pedestrian trajectories, leveraging their natural capacity to capture dynamic motion evolution, such as position shifts and velocity changes, through sequential models like LSTMs (Hochreiter & Schmidhuber, 1997) or Transformers (Vaswani et al., 2017). These approaches excel at modeling local continuity and short-term motion trends, forming the backbone of most state-of-the-art systems.

However, time-domain modeling alone overlooks a crucial aspect: pedestrian motion also exhibits strong regularity in the frequency domain, as revealed by recent studies (Wong et al., 2022; Zhang et al., 2023). Frequency components compactly encode periodic patterns (e.g., gait cycles) and energy distributions, filtering noise while highlighting long-range dependencies that are often obscured in raw time-series data (Wong et al., 2023). Despite these advantages, joint modeling of time and frequency domains remains underexplored.

Motivated by this gap, we initially experimented with a straightforward fusion approach within a dual-branch Transformer framework. Time-domain trajectories and their frequency projections were separately embedded and combined through concatenation. The fused features then served as input tokens of Transformer to predict future trajectory. While this yielded modest improvements, the gains were inconsistent, failing to fully exploit cross-domain synergy. We observe that the fundamental limitation lies in the trajectory representation itself. As illustrated in Figure 1, existing

methods predominantly adopt either grid-based representation (Phan-Minh et al., 2020; Gao et al., 2020; Guo et al., 2022) or point-based representation (Wong et al., 2024; Salzmann et al., 2020; Alahi et al., 2016). Figure 1 (a) divides the environment into fixed cells, leading to loss of precision in representing continuous trajectories. Figure 1 (b) treats trajectory as discrete sequences of coordinates, losing holistic semantics. Neither paradigm can simultaneously capture hierarchical local dynamics and comprehensive global semantics, thereby limiting the predictive capability of subsequent trajectory models. This motivates our rethinking of trajectory representation as structured, semantically meaningful patches.

Drawing inspiration from patch-based representations, which have demonstrated unique advantages in unifying local and global features in computer vision (Liu et al., 2021; Feichtenhofer et al., 2022; Dosovitskiy et al., 2021) and time-series analysis (Brown et al., 2020; Nie et al., 2023; Tang & Zhang, 2025). By adaptively segmenting trajectories into multi-scale spatiotemporal patches where each represents a semantically cohesive motion segment (e.g., a "stepping stride" or "waiting pause"), we enable hierarchical feature learning that preserves local details while modeling long-range dependencies.

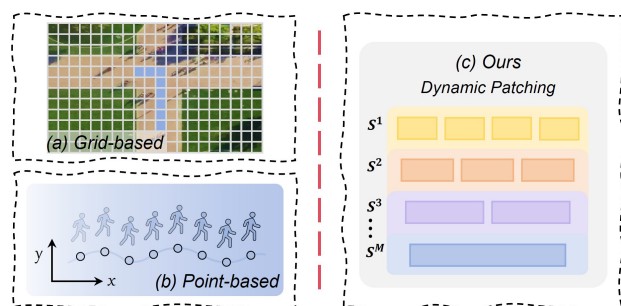

Figure 1: Comparison between existing (a) grid-based methods, and (b) point-based methods for multi-modal trajectory prediction. The former is limited by fixed grid resolution, while the latter loses holistic motion semantics. Our method (c) introduces a dynamic patch mechanism to capture hierarchical local dynamics and comprehensive global semantics.

In this paper, we present **PatchTraj**, a novel dynamic patch-based trajectory prediction framework that unifies time-frequency modeling through dynamic patches. Instead of fixed-length segments, we introduce dynamic patch mechanism, where learns to group trajectory points into semantically meaningful patches based on motion dynamics. Each patch is processed by the expert-inspired embedding layers, followed by hierarchical feature aggregation through a DPAttn module that adaptively samples and fuses motion features across multiple scales. This deformable fusion mechanism dynamically aligns semantically related motion patterns while preserving fine-grained details. The framework processes input trajectories through two complementary branches: a time-domain branch that retains the original sequences, and a frequency-domain branch that derives spectral representations via Discrete Cosine Transform, thereby enabling joint modeling of temporal dynamics and periodic patterns. Additionally, the two branches interact via cross-modal attention, where time-domain queries attend to frequency-domain keys/values to enhance motion semantics. The unified representation is then processed by a Transformer encoder-decoder for autoregressive future trajectory prediction. The main contributions of this paper are summarized as follows:

- We propose a novel dual-branch Transformer-based framework, termed PatchTraj, with time-frequency hybridization for noise-robust trajectory modeling.
- We introduce the first dynamic patch mechanism to adaptively segment trajectories into variable-length temporal and frequency patches to capture heterogeneous motion patterns.
- We further present the scale-aware feature extraction strategy within PatchTraj. The multi-scale patches are processed by adaptive embedding layers, followed by DPAttn aggregator to model both fine-grained and long-range dependencies.
- We experimentally demonstrate that PatchTraj significantly outperforms previous state-of-the-art methods on four real-world datasets, including ETH-UCY, SDD, NBA, and JRDB.

## 2 RELATED WORK

### 2.1 TRAJECTORY PREDICTION

Pedestrian trajectory prediction has become increasingly important for autonomous driving, surveillance, and robotics applications. Early approaches using physical models (Mehran et al., 2009)

and traditional machine learning methods (Morris & Trivedi, 2011; Berndt et al., 2008) established foundations but failed to capture real-world behavioral complexity. The field transformed with deep learning, particularly through RNN-based approaches (Vemula et al., 2018) and LSTM architectures (Alahi et al., 2016) that effectively modeled temporal patterns. Subsequent advances introduced graph-based methods (Huang et al., 2019; Mohamed et al., 2020) to better represent group dynamics and long-range interactions. To handle behavioral uncertainty, researchers developed probabilistic techniques including CVAE (Mangalam et al., 2020; Xu et al., 2022a) and GAN-based approaches (Gupta et al., 2018; Sadeghian et al., 2019) for multi-modal trajectory generation. Most recently, diffusion models (Gu et al., 2022; Mao et al., 2023; Liu et al., 2024b) have emerged as a promising direction, generating trajectories through iterative denoising processes.

Transformers have revolutionized trajectory prediction by overcoming the limitations of sequential models in modeling long-range dependencies and complex interactions. Originally successful in NLP (Vaswani et al., 2017), they were adapted for motion forecasting through self-attention mechanisms that explicitly capture agent relationships and temporal dynamics. Pioneering works like Trajectron++ (Salzmann et al., 2020) combined graph networks with Transformer decoders to generate socially compliant trajectories, while AgentFormer (Yuan et al., 2021) employed a spatio-temporal attention to jointly encode agent histories and future interactions. MART (Lee et al., 2024) is proposed to effectively capture individual and group behaviors via relational transformer mechanisms (Diao & Loynd, 2022).

While most trajectory prediction methods operate solely in the time domain, recent work has revealed valuable periodic patterns in the frequency domain. Several approaches have explored this direction: $V^2$Net (Wong et al., 2022) employs Fourier transforms for hierarchical frequency decomposition, DiffWT (Chen et al., 2025) integrates wavelet transforms with diffusion models, and SpectrumNet (Liu et al., 2024a) encodes frequency features for RNN prediction. However, these methods face critical limitations with static frequency decomposition, complex wavelet processing, and deterministic outputs, preventing them from fully exploiting the complementary relationship between time and frequency domains for optimal prediction performance.

### 2.2 PATCH IN TIME SERIES FORECASTING

Recent advances in time series forecasting have adopted patch-based modeling, inspired by the success of vision transformers (ViTs) (Dosovitskiy et al., 2021) in computer vision, where images are split into non-overlapping patches for efficient self-attention computation. This approach addresses key limitations of RNNs and CNNs, such as limited receptive fields and high computational cost for long sequences. Unlike traditional methods that process time series point-wise or sliding-window-based (e.g., ARIMA, LSTMs), patch-based approaches segment time series into localized temporal patches, enabling efficient long-range dependency modeling and hierarchical feature learning.

The PatchTST model (Nie et al., 2023) pioneered the use of fixed-length patches in time series forecasting, demonstrating improved accuracy through local trend capture and efficient attention computation. Subsequent work has expanded this paradigm in several directions: TimesNet (Wu et al., 2023) employs FFT-based multi-scale patching to identify periodic patterns across resolutions, while FiLM (Zhou et al., 2022a) incorporates patch-level contrastive learning for cross-dataset generalization. To address computational complexity, methods like Informer (Zhou et al., 2021) and FED-former (Zhou et al., 2022b) combine patching with sparse attention mechanisms. PatchMLP (Tang & Zhang, 2025) utilize the MLPs to model interactions among cross variable, which achieves superior performance over existing Transformer models. While these advances establish patch-based modeling as effective for general time series analysis, their potential for trajectory prediction, particularly in capturing the spatiotemporal dynamics of pedestrian motion, remains unexplored.

## 3 METHOD

### 3.1 PROBLEM FORMULATION

Formally, the problem of pedestrian trajectory prediction can be formulated in the temporal domain as follows. For a given pedestrian $i$ in a scene, the observed trajectory over a historical time horizon $t = 1, ..., T_{obs}$ is represented as: $\mathbf{X}^i = [\mathbf{x}_1^i, \mathbf{x}_2^i, ..., \mathbf{x}_{T_{obs}}^i] \in \mathbb{R}^{T_{obs} \times d}$, where $\mathbf{x}_t^i = (x_t^i, y_t^i)$ denotes the 2D spatial coordinates (or $d$-dimensional state, e.g., including velocity, acceleration) at time $t$.

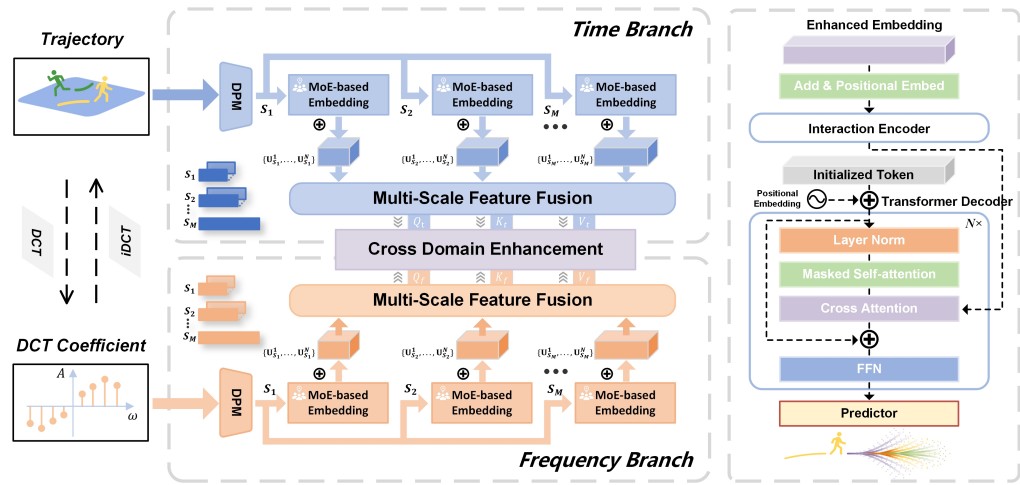

Figure 2: The overall architecture of **PatchTraj**, which is a dual-branch trajectory prediction framework integrating time-domain and frequency-domain processing. The raw time sequences can be transferred into spectral components via DCT. The dynamic patch mechanism is designed to capture multi-granularity motion patterns. Each patch is then processed by expert-inspired embedding layers, and a DPAttn module is utilized to hierarchically aggregate multi-scale features. The outputs of dual branch are fused via cross domain enhancement. Finally, a Transformer encoder-decoder models interactions and predicts future trajectories.

For multi-agent settings, the input includes trajectories of all pedestrians: $\mathbf{X} = \{\mathbf{X}^1, \mathbf{X}^2, ..., \mathbf{X}^N\}$, where $N$ is the number of pedestrians. The goal is to predict the future trajectory distribution over a prediction horizon $t = T_{obs} + 1, ..., T_{pred}$: $p(\mathbf{Y}^i|\mathbf{X}^i, \mathbf{X}^{-i}, \mathbf{C})$, where $\mathbf{Y}^i = [\mathbf{y}^i_{T_{obs}+1}, ..., \mathbf{y}^i_{T_{pred}}] \in \mathbb{R}^{T_{pred} \times d}$ is the future trajectory. $\mathbf{X}^{-i}$ denotes the observed trajectories of other pedestrians (for social interaction modeling) and $\mathbf{C}$ represents contextual information.

We also formulate the frequency-domain transformation of time trajectories using the Discrete Cosine Transform (DCT) (Mao et al., 2019) to enable joint time-frequency analysis in our dual-branch framework shown in Figure 2. For a given observed trajectory $\mathbf{X}^i$, we firstly padding the past trajectory $T_{pred}$ times with the last timestep to form a $T$ length sequence, where $T = T_{obs} + T_{pred}$. Then apply DCT along the temporal axis to obtain spectral coefficients $\mathbf{c}^i \in \mathbb{R}^{T \times d}$: $\mathbf{c}^i_n = \mathrm{DCT}(\mathbf{x}^i_t) = \sum_{t=1}^{T} \sqrt{\frac{2}{N}} \mathbf{x}^i_t \cos[\frac{\pi}{2T}(2t-1)(n-1)]$, where $n \in 1, ..., T$ indexes frequency components. We truncate the spectrum to retain the most informative $l \ll T$ coefficients: $\mathbf{c}^i = \mathbf{c}^i_{1:l} \in \mathbb{R}^{l \times d}$. Due to the DCT orthogonal property, the original data can be reconstructed via iDCT operation.

## 3.2 DYNAMIC PATCH MECHANISM

Effective time series modeling requires precise detection of localized patterns and efficient model refinement. Segmenting data into temporal windows offers the model focused snapshots of short-term series behavior, improving its ability to characterize localized patterns and process these detailed features with greater accuracy. Conventional approaches are used to employ static patches for time series embedding, resulting in models that primarily recognize temporal patterns at a fixed scale, overlooking the inherent multi-scale dynamics and intricate relationships present in temporal data.

To capture local information in more detail and to fully understand the spatial-temporal relationships within the observed trajectory, we propose the dynamic patch mechanism (DPM). Compared to static patching, which divides sequences into fixed-length segments, we define a collection of $M$ patch size values as $\mathcal{S} = \{S_1, ..., S_M\}$, adaptively adjusting the patch size based on the different trajectory length. For a particular patch size $S \in \mathcal{S}$, the observed trajectory $\mathbf{X}$ is first divided into $P$ non-overlapping patches (where $P = T_{obs}/S$) by setting the stride equal to the patch size. Note that it is necessary to ensure the patch size $S$ is divisible by the time length $T$. After $M$ times patch division, we obtain a multi-scale patch sets for both the time-domain and frequency-domain branches. We denote the time-domain patch set as $\mathcal{P}_t = \mathbf{X}^t_{P_1}, ..., \mathbf{X}^t_{P_M}$ and the frequency-domain patch set

(derived from the truncated DCT coefficients $\mathbf{c}^i$) as $\mathcal{P}_f$. The dynamic patch mechanism promotes the spatial-temporal learning ability of trajectory with multi-scale representation, adaptively capturing hierarchical local dynamics and comprehensive global semantics.

### 3.3 MULTI-SCALE PATCH EMBEDDING

To effectively process the multi-scale patch representations generated by our dynamic patch mechanism, we propose a Mixture-of-Experts (Jacobs et al., 1991) based embedding architecture (MSPE). This design enables specialized processing of different temporal granularities while maintaining computational efficiency through sparse expert activation.

Specifically, each expert in our MSPE module is designed to handle specific temporal scales, with dedicated projection weights and positional encodings for every patch size $S \in \mathcal{S}$. This specialization allows individual experts to develop optimized feature extraction capabilities for their assigned temporal granularities. A learnable gating network analyzes the global trajectory context to compute routing weights:

$$\mathbf{G} = \text{Softmax}(\text{MLP}(\text{Flatten}(\mathbf{X}))),$$

where $\mathbf{G} \in \mathbb{R}^{B \times N \times M}$ contains the gating weights for $B$ sequences, $N$ experts, and $M$ patch scales.

For each input sequence, only top-$k$ experts are activated per patch scale, implementing conditional computation:

$$\mathbf{U}_m = \sum\nolimits_{n=1}^{N} \mathbb{I}(n \in \text{TopK}(\mathbf{G}:,n,m)) \cdot \mathbf{W}_n^m(\mathbf{X}_{P_m}),$$

where $\mathbf{W}_n^m$ denotes expert $n$ processing patches of size $S_m$, $\mathbb{I}(\cdot)$ uses 0 or 1 to indicate whether expert $n$ is selected. Due to the expert performance at different scales, we aggregate each expert's learned features along the different scale to construct a comprehensive multi-scale representation.

Our MSPE module elegantly handles multi-scale trajectory patches through a combination of dynamic gating and specialized expert processing. The system intelligently routes different patch scales to appropriate experts using learned attention weights, while maintaining scale-specific projections and positional encodings to preserve unique characteristics at each granularity level.

### 3.4 MULTI-SCALE FEATURE FUSION

When it comes to multi-scale feature fusion, Feature Pyramid Network (FPN) (Lin et al., 2017) is one of the most commonly used methods. We have tried to adopt a FPN to aggregate multi-scale patch features in a top-down manner. While effective in preserving coarse-to-fine structures, this approach fuses features in a uniform and dense fashion, which may fail to capture fine-grained correspondences across scales. In particular, trajectory patches derived from different temporal resolutions or frequency bands often represent semantically aligned motion patterns. Fixed-resolution fusion may dilute such semantics.

To overcome these limitations, we introduce DPAttn, a novel and effective deformable multi-scale attention mechanism specifically designed for trajectory patches. Instead of fusing all scales equally, DPAttn allows each query patch to adaptively sample a sparse set of informative positions across scales, guided by learned offsets and attention weights.

Formally, let $\{\mathbf{U}_1, ..., \mathbf{U}_M\}$ denote aggregated patch features at $M$ different scales. For a query patch embedding $q \in \mathbb{R}^D$, DPAttn aggregates cross-scale information as

$$\text{DPAttn}(q, \{\mathbf{U}_m\}) = \sum\nolimits_{m=1}^{M} \sum\nolimits_{p=1}^{P} A_{m,p}(q) \cdot \mathbf{U}_m(z_{m,p} + \Delta z_{m,p}(q)),$$

where $z_{m,p}$ is the reference location of the $p$-th sampled patch at scale $m$, $\Delta z_{m,p}(q)$ is a query-dependent learnable offset, and $A_{m,p}(q)$ is the normalized attention weight. Linear interpolation is applied when offsets fall between discrete patch indices, ensuring smooth aggregation.

To guarantee dimensional consistency during fusion, the aggregated outputs are projected back to a common embedding space through linear transformation. The resulting representations from the temporal and frequency branches, denoted as $\mathcal{F}_t$ and $\mathcal{F}_f$, are enriched with dynamically aligned cross-scale information while retaining fine-grained details. This deformable fusion mechanism equips the model with the ability to adaptively focus on salient motion cues across multiple resolutions, thereby capturing human spatiotemporal dynamics.

### 3.5 CROSS DOMAIN ENHANCEMENT

While Figure 2 illustrates our dual-branch architecture for processing time and frequency inputs, we further bridge these modalities by interacting final fused features $\mathcal{F}_t$ and $\mathcal{F}_f$ through the cross-attention mechanism. To compensate for the positional insensitivity of patch-based representations, we augment both feature streams with learnable positional encodings. The enhancement module then establishes bidirectional interaction via:

**Temporal-to-frequency attention** enables time features to dynamically attend to relevant frequency components. The temporal patches serve as queries to retrieve complementary spectral information through dot-product attention: $\hat{\mathcal{F}}_t = \text{Attention}(Q_t, K_f, V_f) = \text{softmax}(\frac{Q_t K_f^T}{\sqrt{d_k}})V_f$.

**Frequency-to-temporal attention** conversely allows spectral features to assimilate critical temporal patterns. This reverse attention flow helps localize periodic motion characteristics in time domain: $\hat{\mathcal{F}}_f = \text{Attention}(Q_f, K_t, V_t) = \text{softmax}(\frac{Q_f K_t^T}{\sqrt{d_k}})V_t$.

Residual connection preserves original modality-specific features while incorporating cross-domain enhancements through element-wise addition: $\mathcal{F}_t' = \mathcal{F}_t + \hat{\mathcal{F}}_t$ and $\mathcal{F}_f' = \mathcal{F}_f + \hat{\mathcal{F}}_f$. The concatenated $[\mathcal{F}_t'; \mathcal{F}_f']$ maintain both domains' distinctive properties while capturing their latent correlations.

### 3.6 INTERACTION ENCODER

Our framework employs a Transformer encoder with skip connections (Ronneberger et al., 2015) to model fine-grained interactions between pedestrians from fused multi-modal patch embeddings. The raw time-domain trajectories are encoded via MLPs, while modality-aware features are preserved through residual connections to maintain temporal awareness. Within each transformer block, multi-head attention operates on the patch embeddings, capturing both long-range and local dependencies between agents at different time steps. This allows the model to explicitly reason about social interactions by attending to relevant agents and time periods.

### 3.7 TRAJECTORY DECODER

The prediction pipeline employs an autoregressive decoder with $N$ transformer blocks to generate future trajectories from encoded representations. We initialize a learnable prediction token $\mathcal{T} \in \mathbb{R}^{T_{pred} \times D}$ as decoder inputs. Each decoder layer performs cross-attention between prediction queries and encoder output features. We use proper masking inside decoder to enforce causality of the decoder output sequence. Final MLP head predicts trajectory coordinates:

$$\hat{\mathbf{Y}} = \text{MLP}(\text{Decoder}(\mathcal{T}^{(N)}, [\mathcal{F}_t'; \mathcal{F}_f'])) \in \mathbb{R}^{K \times T_{pred} \times 2},$$

where output maintains multiple hypotheses ($K$ samples) for multi-modal prediction.

### 3.8 TRAINING CONSTRAINT

**Frequency reconstruction.** To ensure that the predicted trajectories are also consistent with the observed motion patterns in the frequency domain, we introduce a frequency reconstruction loss $\mathcal{L}_{freq}$. Specifically, we apply the iDCT operation to the predicted trajectory $\hat{\mathbf{Y}}$ for each sample to reconstruct its frequency components $\hat{\mathbf{c}} = \text{iDCT}(\hat{\mathbf{Y}}) \in \mathbb{R}^{l \times d}$. We then compute the Mean Squared Error (MSE) between the reconstructed frequency components $\hat{\mathbf{c}}$ and the original low-frequency components $\mathbf{c}$ for the best-matched sample:

$$\mathcal{L}_{freq} = \min_k^K \|\mathbf{c} - \hat{\mathbf{c}}^k\|_2.$$

**Trajectory reconstruction.** We simultaneously optimize *marginal loss* (Gupta et al., 2018) and *joint loss* (Weng et al., 2023) by employing L2-norm among $K$ samples to minimize the distance between the prediction and the ground truth. Please refer to our proof in the appendix for joint training details.

$$\mathcal{L}_{marginal} = \sum_n^N \min_k^K \|\mathbf{Y}_n - \hat{\mathbf{Y}}_n^k\|_2, \quad \mathcal{L}_{joint} = \min_k^K \sum_n^N \|\mathbf{Y}_n - \hat{\mathbf{Y}}_n^k\|_2.$$

**Total loss.** The framework is trained in an end-to-end strategy by combining the triple loss items:

$$\mathcal{L} = \mathcal{L}_{marginal} + \lambda \mathcal{L}_{joint} + \mathcal{L}_{freq}.$$

## 4 EXPERIMENTS

### 4.1 DATASETS

The model is trained and evaluated on four publicly available datasets for trajectory prediction: ETH-UCY (Pellegrini et al., 2010), Stanford Drone Dataset (SDD) (Robicquet et al., 2016), NBA SportVU Dataset (NBA) (Yue et al., 2014), and the JackRabbot Dataset and Benchmark (JRDB) (Martin-Martin et al., 2021). **ETH-UCY** dataset comprises five distinct subsets: ETH, HOTEL, UNIV, ZARA1 and ZARA2. We follow the leave-one-out approach from (Gupta et al., 2018) with four subsets for training-validation and the remaining subset for testing. We utilize the standard dataset splits to predict the future 12 frames (4.8s) with 8 frames observations (3.2s). **SDD** dataset captures large-scale pedestrian behaviors on a campus from a bird's-eye perspective. The dataset is split into predicting the future 12 frames (4.8s) with 8 frames observations (3.2s). **NBA** dataset tracks the movements of 10 basketball players and a ball during NBA games in 2015-2016 season. The movements exhibit strong purpose fulness and complex variety, which increases difficulties compared to the pedestrian datasets. Followed by (Mao et al., 2023), we predict the future 20 frames (4.0s) based on 10 frames (2.0s) history. **JRDB** is a large-scale egocentric dataset recorded by a social robot in indoor and outdoor scenarios with stationary and moving behaviors. We follow the deterministic prediction split in (Saadatnejad et al., 2024) and the multi-modal prediction split in (Fang et al., 2025), which predicts the future 12 frames (4.8s) over the observed 9 frames (3.6s).

### 4.2 IMPLEMENTATION DETAILS

We represent each trajectory as a 6-dimensional vector ($d = 6$) combining absolute position, relative displacement, and velocity. For frequency-domain processing, we employ truncated DCT operations retaining the first $l$ coefficients, with dataset-specific values: $l = 8$ (ETH-UCY/SDD), $l = 10$ (NBA), and $l = 9$ (JRDB). This maintains dimensional consistency between time and frequency branches while preserving key spectral components. To obtain non-overlapping patches, we set a list of dynamic patch size $\mathcal{S} = \{2, 4, 8\}$ for ETH-UCY/SDD, $\mathcal{S} = \{2, 5, 10\}$ for NBA, $\mathcal{S} = \{1, 3, 9\}$ for JRDB according to the history length. Four experts are used in the MSPE module and top-2 experts are selected. We employ $L = 4$ layers with $H = 4$ attention heads in each Transformer-based sub-network, and the dimension of hidden state and patch embedding is set $D = 256$. The model is implemented in PyTorch (Paszke et al., 2019) and optimized using AdamW (Loshchilov & Hutter, 2019) with an initial learning rate of $1 \times 10^{-3}$ which halve every 10 epochs, and trained for 100 epochs on a single 4090 GPU. For the loss coefficient weight, we empirically set $\lambda = 0.5$.

### 4.3 EVALUATION METRICS

We adopt two widely reported metrics to evaluate the trajectory prediction performance: the Average Displacement Error (ADE) and the Final Displacement Error (FDE) (Pellegrini et al., 2009). ADE and FDE measures the accuracy between the ground truth and predicted trajectory over all timestep and the last timestep. Inspired by (Gupta et al., 2018), we adopt multi-modal prediction and generate $K$ samples for each trajectory. Consequently, minADE$_K$ and minFDE$_K$ are reported in our evaluation results.

### 4.4 QUANTITATIVE RESULTS

*JRDB.* Table 1-(a) compares our method with 5 prominent methods on the JRDB dataset. For fair comparison with deterministic baselines, we configure PatchTraj to generate single-sample predictions. Our method establishes new state-of-the-art performance, demonstrating significant improvements of 23.1% in ADE and 16.7% in FDE over the previous best approach NMRF.

*NBA.* Table 1-(b) compares our method with 5 prominent methods on the NBA dataset. Results show that our PatchTraj reduces ADE/FDE from 0.75/0.97 to 0.68/0.94 compared with NMRF in

Table 1: Quantitative comparison results on (a) JRDB, (b) NBA, (c) SDD and (d) ETH-UCY datasets. **ADE** and **FDE** are reported on (a) for deterministic prediction, **minADE$_{20}$** and **minFDE$_{20}$** are reported on (b), (c), (d) for multi-modal prediction. **Bold** and underlined fonts represent the best and second-best results, respectively (lower values are better).

| **(a) JRDB Dataset** ($K = 1$) | | | | | |
|---|---|---|---|---|---|
| Time | Trajectron++ (Salzmann et al., 2020) | EqMotion (Xu et al., 2023) | Social-Trans (Saadatnejad et al., 2024) | EmLoco (Taketsugu et al., 2025) | NMRF (Fang et al., 2025) | Ours |
| 4.8s | 0.40/0.78 | 0.42/0.78 | 0.40/0.77 | 0.37/0.72 | 0.26/0.48 | **0.20/0.40** |

| **(b) NBA Dataset** ($K = 20$) | | | | | |
|---|---|---|---|---|---|
| Time | Trajectron++ (Salzmann et al., 2020) | GroupNet (Xu et al., 2022a) | LED (Mao et al., 2023) | MART (Lee et al., 2024) | NMRF (Fang et al., 2025) | Ours |
| 4.0s | 1.15/1.57 | 0.96/1.30 | 0.81/1.10 | 0.73/**0.90** | 0.75/0.97 | **0.68**/0.94 |

| **(c) SDD Dataset** ($K = 20$) | | | | | |
|---|---|---|---|---|---|
| Time | V$^2$Net (Wong et al., 2022) | LED (Mao et al., 2023) | MGF (Chen et al., 2024) | MART (Lee et al., 2024) | NMRF (Fang et al., 2025) | Ours |
| 4.8s | 7.12/11.39 | 8.48/11.66 | 7.74/12.07 | 7.43/11.82 | 7.20/11.29 | **6.58/11.14** |

| **(d) ETH-UCY Dataset** ($K = 20$) | | | | | |
|---|---|---|---|---|---|
| Subset | AgentFormer (Yuan et al., 2021) | LED (Mao et al., 2023) | MART (Lee et al., 2024) | MoFlow (Fu et al., 2025) | NMRF (Fang et al., 2025) | Ours |
| ETH | 0.45/0.75 | 0.39/0.58 | 0.35/0.47 | 0.40/0.57 | **0.26/0.37** | 0.30/0.48 |
| HOTEL | 0.14/0.22 | 0.11/0.17 | 0.14/0.22 | 0.11/0.17 | 0.11/0.17 | **0.10/0.16** |
| UNIV | 0.25/0.45 | 0.26/0.43 | 0.25/0.45 | **0.23/0.39** | 0.28/0.49 | **0.23**/0.45 |
| ZARA1 | 0.18/0.30 | 0.18/**0.26** | 0.17/0.29 | 0.15/**0.26** | 0.17/0.30 | **0.14**/0.27 |
| ZARA2 | 0.14/0.24 | 0.13/0.22 | 0.13/0.22 | 0.12/0.22 | 0.14/0.25 | **0.10/0.19** |
| AVG | 0.23/0.39 | 0.21/0.33 | 0.21/0.33 | 0.20/0.32 | 0.19/0.32 | **0.17/0.31** |

total 4 seconds, improving performance by 9.3% and 3.1%, respectively. We achieve the best performance on ADE, but the absence of explicit intention-aware modeling fundamentally limits FDE performance.

*SDD.* Table 1-(c) compares our method with 5 prominent methods on the SDD dataset, with a sample number limit of $K = 20$ for all comparative methods. Our method maintains state-of-the-art accuracy to the best-of-20 samples. Specifically, we surpass outstanding spectral-based V$^2$Net by improving ADE/FDE from 7.12/11.39 to 6.58/11.14. Compared to the latest state-of-the-art method NMRF, PatchTraj reduces ADE from 7.20 to 6.58, achieving 8.6% large improvement. A reasonable interpretation is that we propose the dynamic patch mechanism to capture local motion patterns with trajectory segments.

*ETH-UCY.* As demonstrated in Table 1-(d), PatchTraj achieves superior performance compared to 8 state-of-the-art methods across nearly all ETH-UCY subsets. Our approach shows particular advantages over Transformer-based baselines (AgentFormer, MART) through its dynamic patch mechanism, which effectively captures multi-scale motion patterns. Notably, PatchTraj outperforms the current state-of-the-art NMRF by significant margins, reducing ADE by 10.5% and FDE by 3.1%. These consistent improvements across diverse scenarios validate the effectiveness of our method in pedestrian trajectory prediction.

## 4.5 QUALITATIVE RESULTS

To have a more intuitive awareness of our PatchTraj performance, we visualize the overall distribution and optimal outcome from 20 predictions in Figure 3. The results show that the predicted trajectories greatly align with the ground truth, demonstrating the effectiveness of our method. Please refer to the more visualizations in the appendix.

## 4.6 ABLATION STUDIES

We further perform extensive ablation studies on NBA, SDD and ETH-UCY datasets illustrated in Table 2 to investigate the contribution of key components in our method, with sample number fixed at

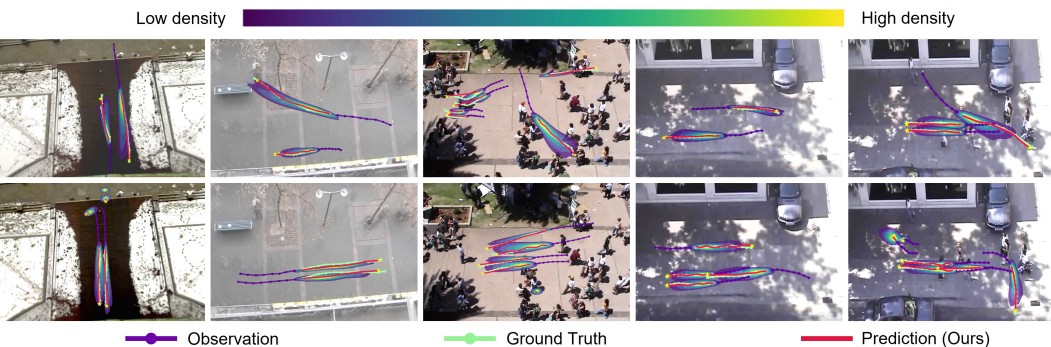

Figure 3: Visualization results on the ETH-UCY dataset, including heatmaps of the overall distribution of predicted samples and best-of-20 predictions of our method.

Table 2: Ablations to study the contribution of key technical components on model performance. Average ADE and FDE are reported on NBA, SDD and ETH-UCY datasets. "T" denotes the time and "F" denotes the frequency. "DPM" indicates dynamic patch mechanism, and "MSPE" means expert-inspired multi-scale patch embedding. "MSFF" means multi-scale feature fusion with DPAttn, and "CDE" indicates the cross domain enhancement. $K$ is the number of multi-modal prediction.

| T-branch | F-branch | DPM | MSPE | MSFF | CDE | N-Sample | NBA | | SDD | | ETH-UCY | |
|---|---|---|---|---|---|---|---|---|---|---|---|---|
| | | | | | | $K$ | ADE | FDE | ADE | FDE | ADE | FDE |
| ✓ | - | - | - | - | - | 20 | 1.02 | 1.43 | 8.03 | 15.30 | 0.32 | 0.55 |
| ✓ | ✓ | - | - | - | - | 20 | 0.96 | 1.38 | 7.68 | 14.24 | 0.30 | 0.52 |
| ✓ | ✓ | ✓ | - | - | - | 20 | 0.82 | 1.10 | 7.05 | 12.57 | 0.24 | 0.48 |
| ✓ | ✓ | ✓ | ✓ | - | - | 20 | 0.77 | 1.08 | 6.89 | 11.71 | 0.22 | 0.42 |
| ✓ | ✓ | ✓ | ✓ | ✓ | - | 20 | 0.71 | 1.00 | 6.68 | 11.64 | 0.19 | 0.36 |
| ✓ | ✓ | ✓ | ✓ | ✓ | ✓ | 20 | **0.68** | **0.94** | **6.58** | **11.14** | **0.17** | **0.31** |

$K = 20$ for consistency. As shown in the first row of Table 2, the vanilla Transformer with separate time-domain inputs establishes our baseline performance. The second line suggests that constructing a dual-branch architecture to simultaneously integrate time and frequency trajectory information can improve prediction performance. The third line demonstrates that our proposed dynamic patch mechanism enhances time-frequency representation through local dynamic and global dependency modeling. The two key components of "MSPE" and "MSFF" further improve model performance. The "CDE" module effectively establishes cross-modal relationships, enabling synergistic fusion of heterogeneous features. The cumulative improvements demonstrate that each component contributes uniquely to our state-of-the-art performance, with the complete framework achieving a large enhancement over the baseline. Please refer to the more ablations in the appendix.

## 5 CONCLUSION

In this paper, we present PatchTraj, a novel dynamic patch-based trajectory prediction framework that unifies time-domain and frequency-domain representations. Our approach addresses the limitations of existing methods by introducing a dynamic patch mechanism to capture multi-granularity motion patterns, an expert-inspired embedding layer for scale-aware feature extraction, and a hierarchical feature fusion module DPAttn to balance fine-grained details with long-range dependencies. The cross-domain enhancement further leverages the complementary strengths of time and frequency representations, enabling robust and noise-resistant trajectory modeling. Extensive experiments on four real-world datasets (ETH-UCY, SDD, NBA, and JRDB) have demonstrated that PatchTraj achieves state-of-the-art performance, outperforming existing methods in both accuracy and robustness. Ablation studies validate the contributions of each key component, highlighting the effectiveness of our design choices.

ETHICS STATEMENT

This research leverages only publicly available trajectory prediction datasets, including ETH-UCY, Stanford Drone Dataset (SDD), NBA SportVU, and JRDB. No personally identifiable or sensitive information is used, and all datasets were originally collected with appropriate permissions and have been widely adopted in prior research. The study does not involve direct interaction with human subjects, nor does it raise issues of discrimination, bias, or privacy leakage. The proposed methodology is intended to advance trajectory prediction for autonomous driving and robotics, and it does not generate harmful or malicious applications. We have carefully ensured that our work adheres to the ICLR Code of Ethics.

REPRODUCIBILITY STATEMENT

We have taken multiple steps to ensure reproducibility of our results. All implementation details, including model architecture, training strategy, and hyperparameters, are provided in Section 3 (Method) and Section 4.2 (Implementation Details) of the main text, with additional proofs and ablation studies presented in the Appendix. We explicitly describe dataset splits, temporal settings, and evaluation metrics in Section 4.1 and Appendix A.3–A.4. Furthermore, our model is implemented in PyTorch, and we will release the anonymized source code and instructions for dataset preprocessing as supplementary material. Together, these efforts make it straightforward for other researchers to replicate and build upon our work.

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

# A APPENDIX

## A.1 MODEL ARCHITECTURE PARAMETERS

Table 3 details the architectural parameters of the PatchTraj model. It is noted that the majority of the parameters derive from the Transformer module, specifically due to the number of layers and attention heads. We have investigated alternative lightweight architectures, such as LSTMs and RNNs, and observed a significant degradation in performance. To achieve a balance between accuracy and computational efficiency, we employed a relatively small number of transformer layers and heads.

| Parameter | Value | Description |
|---|---|---|
| *Basic Model Parameters* | | |
| input_dim | 6 | augmented trajectory inputs |
| output_dim | 2 | absolute position (x, y) |
| obs_len | 8 | length of observation |
| pred_len | 12 | length of future |
| $l$ | 8 | length of truncated spectral components |
| num_sample | 20 | number of trajectory sample |
| padding | 'LastFrame' | padding style |
| *Dynamic Patch Mechanism* | | |
| dynamic_patch | True | whether use dynamic patch |
| patch_list | [2, 4, 8] | a list of patch length |
| *Multi-scale Patch Embedding* | | |
| patch_embed | 256 | patch embedding dimension |
| num_experts | 4 | number of expert |
| top_k | 2 | number of activated experts |
| *Cross Domain Enhancement* | | |
| latent_dims | 256 | dimension of latent feature |
| num_heads | 4 | number of attention heads |
| *Sub Transformer Module* | | |
| query_dim | 256 | dimension of input query |
| num_layers | 4 | number of transformer layer |
| num_heads | 4 | number of attention heads |
| dropout | 0.2 | dropout rate for regularization |
| *Total number of model parameters: 7.63M* | | |

Table 3: Model architecture parameters for PatchTraj.

## A.2 FURTHER DISCUSSION ABOUT TRAINING CONSTRAINT

Our framework is trained in an end-to-end strategy by simultaneously optimizing *marginal loss* (Gupta et al., 2018) and *joint loss* (Weng et al., 2023) to minimize the distance between the prediction and the ground truth.

$$\mathcal{L}_{marginal} = \sum_n^N \min_k^K \|\mathbf{Y}_n - \hat{\mathbf{Y}}_n^k\|_2,$$

$$\mathcal{L}_{joint} = \min_k^K \sum_n^N \|\mathbf{Y}_n - \hat{\mathbf{Y}}_n^k\|_2,.$$

In paper (Weng et al., 2023), the author specified that multi-modal trajectory prediction methods are usually trained with marginal loss and evaluated on marginal metrics ADE / FDE, which fail to capture the joint performance of multiple interacting agents. To compensate for this limitation, the author added an extra joint-loss term with marginal loss to jointly train previous state-of-the-art methods. The results show that performances are improved on ETH-UCY and SDD datasets. A reasonable explanation provided by the author is that the joint prediction problem is inherently more

complex than the marginal prediction problem due to having to optimize for the joint performance of multiple agents rather than individual agents independently. Here, we assume that **optimizing for joint loss is harder than marginal loss, and convergence on the joint loss can help the marginal loss converge**. Our proof lists as follows:

$$\text{let } \triangle \mathbf{Y}_n^k = \mathbf{Y}_n^k - \hat{\mathbf{Y}}_n^k,$$

$$\text{for } \forall k, \mathcal{L}_{marginal} - \sum_n^N \|\mathbf{Y}_{t,n}^k - \hat{\mathbf{Y}}_{t,n}^k\|_2$$

$$= \sum_n^N \min_k^K \|\triangle \mathbf{Y}_n^k\|_2 - \sum_n^N \|\triangle \mathbf{Y}_n^k\|_2$$

$$= \sum_n^N \left( \min_k^K \|\triangle \mathbf{Y}_n^k\|_2 - \|\triangle \mathbf{Y}_n^k\|_2 \right)$$

$$\because \min_k^K \|\triangle \mathbf{Y}_n^k\|_2 - \|\triangle \mathbf{Y}_n^k\|_2 \le 0,$$

$$\therefore \mathcal{L}_{marginal} - \sum_n^N \|\mathbf{Y}_{t,n}^k - \hat{\mathbf{Y}}_{t,n}^k\|_2 \le 0,$$

$$\therefore \mathcal{L}_{marginal} - \min_k^K \sum_n^N \|\mathbf{Y}_{t,n}^k - \hat{\mathbf{Y}}_{t,n}^k\|_2 \le 0,$$

$$\therefore \mathcal{L}_{marginal} - \mathcal{L}_{joint} \le 0,$$

$$\therefore \mathcal{L}_{marginal} \le \mathcal{L}_{joint}.$$

### A.3 DETAIL INFORMATION OF DATASETS

Table 4 shows the details of our implemented datasets.

Table 4: Various temporal configurations during data acquisition across trajectory datasets.

| Dataset | Past (s) | Future (s) | Frequency (Hz) | Training size |
|---------|----------|------------|----------------|---------------|
| ETH-UCY | 3.2 | 4.8 | 2.5 | 36497 |
| SDD | 3.2 | 4.8 | 2.5 | 17970 |
| NBA | 2.0 | 4.0 | 5.0 | 40000 |
| JRDB | 3.6 | 4.8 | 2.5 | 92803 |

### A.4 FORMULATION OF EVALUATION METRICS

We explicitly display the calculation details of evaluation metrics which not present in our paper, as following:

$$ADE = \frac{1}{TN} \sum_{n=1}^N \min_{k=1}^K \sum_{t=1}^T \|\mathbf{y}_{t,n}^k - \hat{\mathbf{y}}_{t,n}^k\|_2,$$

$$FDE = \frac{1}{N} \sum_{n=1}^N \min_{k=1}^K \|\mathbf{y}_{t_{pred},n}^k - \hat{\mathbf{y}}_{t_{pred},n}^k\|_2.$$

In our paper, we mainly evaluate and report with marginal metrics of ADE and FDE. Additionally, we introduce the joint metrics of JADE and JFDE to adapt for joint loss term, comparing the impact of two different losses on model performance.

$$JADE = \frac{1}{TN} \min_{k=1}^K \sum_{n=1}^N \sum_{t=1}^T \|\mathbf{y}_{t,n}^k - \hat{\mathbf{y}}_{t,n}^k\|_2,$$

$$JFDE = \frac{1}{N} \min_{k=1}^K \sum_{n=1}^N \|\mathbf{y}_{t_{pred},n}^k - \hat{\mathbf{Y}}_{t_{pred},n}^k\|_2.$$

Table 5: Quantitative comparison results on (a) ETH-UCY and (b) SDD datasets. **minJADE**$_{20}$ and **minJFDE**$_{20}$ are reported for multi-modal prediction ($K = 20$). **Bold** and underlined fonts represent the best and second-best results, respectively (lower values are better).

| | | | | **(d) ETH-UCY Dataset** ($K = 20$) | | | |
|---|---|---|---|---|---|---|---|
| Subset | Trajectron++ (Salzmann et al., 2020) | AgentFormer (Yuan et al., 2021) | V$^2$Net (Wong et al., 2022) | MemoNet (Xu et al., 2022b) | MGF (Chen et al., 2024) | NMRF (Fang et al., 2025) | Ours |
| ETH | 0.73/1.30 | 0.48/0.79 | 0.56/0.78 | 0.50/0.86 | 1.65/3.50 | 0.54/1.04 | **0.40/0.72** |
| HOTEL | 0.24/0.42 | 0.24/0.46 | 0.20/**0.33** | 0.22/0.42 | 0.71/1.57 | 0.26/0.48 | **0.17**/0.34 |
| UNIV | 0.61/1.32 | 0.62/1.31 | 0.65/1.47 | 0.69/1.47 | 0.89/1.99 | 0.64/1.31 | **0.47/1.09** |
| ZARA1 | 0.36/0.71 | **0.29/0.56** | 0.33/0.65 | 0.35/0.72 | 0.74/1.56 | 0.39/0.82 | 0.31/0.73 |
| ZARA2 | 0.29/0.63 | 0.30/0.62 | 0.30/0.60 | 0.39/0.86 | 0.71/1.60 | 0.33/0.70 | **0.24/0.58** |
| AVG | 0.45/0.87 | 0.38/0.75 | 0.41/0.73 | 0.43/0.87 | 0.94/2.04 | 0.43/0.87 | **0.32/0.69** |
| | | | | **(c) SDD Dataset** ($K = 20$) | | | |
| Time | Trajectron++ (Salzmann et al., 2020) | AgentFormer (Yuan et al., 2021) | V$^2$Net (Wong et al., 2022) | MemoNet (Xu et al., 2022b) | MGF (Chen et al., 2024) | NMRF (Fang et al., 2025) | Ours |
| 4.8s | 21.36/44.21 | 19.45/42.04 | 17.16/36.82 | 18.81/40.56 | 35.02/71.48 | 15.99/32.33 | **14.37/31.83** |

Table 6: Multi-modal trajectory prediction comparisons on JRDB dataset. **minADE**$_{20}$/**minFDE**$_{20}$ in meters are reported for the future 12 frames (4.8s).

| Time | LED (Mao et al., 2023) | MART (Lee et al., 2024) | Social-Trans (Saadatnejad et al., 2024) | NMRF (Fang et al., 2025) | Ours |
|---|---|---|---|---|---|
| 1.2s | 0.05/0.07 | 0.06/0.07 | 0.08/0.10 | 0.04/0.05 | **0.02/0.03** |
| 2.4s | 0.09/0.14 | 0.10/0.14 | 0.12/0.16 | 0.08/0.11 | **0.05/0.08** |
| 3.6s | 0.14/0.21 | 0.14/0.20 | 0.17/0.23 | 0.11/0.17 | **0.08/0.13** |
| 4.8s | 0.18/0.28 | 0.18/0.26 | 0.21/0.30 | 0.15/0.23 | **0.11/0.19** |

## A.5 ADDITIONAL QUANTITATIVE RESULTS

***Evaluation with the joint metrics.*** Joint metrics (Table 5) perform about 2x worse across the board as compared to marginal metrics (Table 1), which demonstrates the hypothesis again that marginal metrics are overly optimistic estimations of trajectory prediction performance. Notably, our method not only achieves superior predictions for individual agents across diverse prediction samples but also achieves modeling joint futures and interactions for multi-agent forecasting. In Table 5, we can see that our method outperforms those previously proposed approaches by reducing the values of JADE and JFDE, achieving state-of-the-art performances on both ETH-UCY and SDD datasets. Specifically, compared with the current state-of-the-art method NMRF on ETH-UCY, our Patch-Traj reduces JADE from 0.43 to 0.32 and JFDE from 0.87 to 0.69, achieving a 25.6% and a 20.7% improvement respectively. A reasonable phenomenon appears on the other dataset SDD, with numerous interactions caused by high-density sequences. Our method achieves the best performance.

***Multi-modal prediction performance on JRDB dataset.*** Due to the page limitation, the main text only includes the deterministic result ($K = 1$) on JRDB dataset. We also conduct experiments to compare the multi-modal prediction ($K = 20$), the superior performance presented in Table 6 further investigate the effectiveness of our method. Specifically, compared with the current state-of-the-art method NMRF, our PatchTraj reduces ADE from 0.15 to 0.11 and FDE from 0.23 to 0.19 in total 4.8s, achieving a 26.7% and a 17.4% improvement respectively. Achieving state-of-the-art multi-modal performance on JRDB indicates that our method not only captures diverse and uncertain pedestrian intentions but also generalizes robustly to highly dynamic and cluttered real-world scenes. This highlights the strength of our representation in aligning local dynamics with global structures, making it particularly suitable for downstream applications such as autonomous navigation and human-robot interaction.

## A.6 ADDITIONAL ABLATION STUDIES

***Ablation to training loss.*** The preceding proof has revealed the relationship of marginal loss and joint loss, we conduct ablations to explicitly study the impact of two different losses on model per-

Table 7: Ablations to study the impact of marginal loss $\mathcal{L}_m$ (simplified form) and joint loss $\mathcal{L}_j$ (simplified form) on model performance. Experiments are conducted on ETH-UCY dataset and all metrics are reported on the average of best-of-20 samples.

| Loss | | Avg Metric | | | |
|---|---|---|---|---|---|
| $\mathcal{L}_m$ | $\mathcal{L}_j$ | ADE↓ | FDE↓ | JADE↓ | JFDE↓ |
| ✓ | - | 0.19 | 0.38 | 0.45 | 0.88 |
| - | ✓ | 0.33 | 0.53 | 0.34 | 0.69 |
| ✓ | ✓ | **0.17** | **0.31** | **0.32** | **0.62** |

Table 8: Ablations to study the different methods of frequency domain transformation. Experiments are conducted on ETH-UCY and SDD datasets, and all metrics are reported on the average of best-of-20 samples.

| Method | Time | ETH-UCY | | SDD | |
|---|---|---|---|---|---|
| | | ADE↓ | FDE↓ | ADE↓ | FDE↓ |
| DFT | 0.1ms | 0.18 | 0.35 | 6.69 | 11.63 |
| DWT | 1.5ms | 0.20 | 0.41 | 6.82 | 11.88 |
| DCT | 0.2ms | **0.17** | **0.31** | **6.58** | **11.14** |

Table 9: Ablations to study the different number of experts in MSPE module. Experiments are conducted on ETH-UCY and SDD datasets, and all metrics are reported on the average of best-of-20 samples.

| Expert Number | ETH-UCY | | SDD | |
|---|---|---|---|---|
| | ADE↓ | FDE↓ | ADE↓ | FDE↓ |
| 2 | 0.20 | 0.42 | 6.98 | 12.24 |
| 4 | **0.17** | **0.31** | **6.58** | **11.14** |
| 6 | 0.19 | 0.39 | 6.78 | 11.51 |
| 8 | 0.19 | 0.40 | 6.79 | 11.64 |

formance with the supervised training of $\mathcal{L}_m$ and/or $\mathcal{L}_j$. Intuitively, the joint loss focuses on multiple agent prediction, while the marginal loss cares for individual agent prediction. An interesting result in Table 7 shows that the training incorporated with marginal loss $\mathcal{L}_{marginal}$ and joint loss $\mathcal{L}_{joint}$ leads to an improvement on both ADE/FDE and JADE/JFDE metrics. A reasonable explanation is that training with marginal and joint loss encourages the model to produce accurate predictions for each agent in any of its $K$ samples and to predict realistic trajectories from multi-agent interaction.

***Ablation to frequency transformation.*** We conduct comprehensive ablation studies to evaluate different spectral transformation approaches for trajectory representation shown in Table 8. Note that we retain the same experimental setup except for the spectral transformation method. Discrete Fourier Transform (DFT) transfers time trajectory coordinates into complex numbers containing real and imaginary parts with sine and cosine components, capturing global frequency features. The results in Table 8 achieve the second best performance on both two datasets. Discrete Wavelet Transform (DWT) transfers time trajectories into multi-level coefficients, where we set parameters of 'db1' wavelet and level=2. We see that the performance seriously degrades compared to DFT on both two datasets, meanwhile the transformation consumes a lot of time. As evidenced in Table 8, the Discrete Cosine Transform (DCT) emerges as the superior choice for trajectory transformation in frequency representation, demonstrating DCT keeps a balance between computational efficiency and prediction accuracy.

***Ablation to expert number.*** Our MSPE module elegantly handles multi-scale trajectory patches through a combination of dynamic gating and specialized expert processing. To further explore the impact of expert number on the model performance, we perform ablations on ETH-UCY and SDD datasets in Table 9. The experimental results demonstrate that the optimal performance on both datasets is achieved with 4 experts. However, when the number of experts increases to 6 or

Table 10: Ablations to study the impact of different sample number $K$ on multi-modal prediction performance. Experiments are conducted on ETH-UCY and SDD datasets, and all metrics are reported on the average of best-of-k samples. **Bold** font represents the best result, and the lower the better.

| Dataset | Metric | $K$=2 | $K$=4 | $K$=8 | $K$=10 | $K$=12 | $K$=16 | $K$=20 |
|---------|--------|-------|-------|-------|--------|--------|--------|--------|
| ETH-UCY | ADE | 0.348 | 0.252 | 0.224 | 0.208 | 0.194 | 0.188 | **0.174** |
|         | FDE | 0.684 | 0.452 | 0.404 | 0.382 | 0.362 | 0.336 | **0.310** |
| SDD     | ADE | 11.721 | 9.670 | 8.006 | 8.301 | 6.934 | 6.706 | **6.581** |
|         | FDE | 24.243 | 19.213 | 15.020 | 15.053 | 12.545 | 11.638 | **11.142** |

8, both datasets exhibit consistent performance degradation. This finding suggests that employing a moderate number of experts can effectively capture multi-scale trajectory features while maintaining computational efficiency.

***Ablation to sample number.*** This experiment intends to clarify the impact of different sample numbers $K$ on multi-modal prediction. The value of $K$ means the number of generated samples from the same distribution in the inference phase. With $K$ gradually increasing, shown in Table 10, the values of ADE&FDE are correspondingly decreasing, which presents a positive correlation between both datasets. The higher the $K$ value, the better the evaluation performance. We follow the standard setting $K = 20$ in multi-modal pedestrian trajectory prediction.

## A.7 MORE VISUALIZATION RESULTS

Figure 4 compares our method's optimal prediction from 20 trajectory samples against the state-of-the-art method NMRF (Fang et al., 2025). The results particularly highlight that PatchTraj's predicted trajectories align best with the ground truth and maintain consistent path fidelity across all scenarios (ETH, HOTEL, UNIV, ZARA1 and ZARA2), demonstrating the superiority of our proposed method.

Visualization results on ETH-UCY dataset in Figure 5 further illustrate that our method preserves inherent motion patterns of pedestrians while retaining the multi-modal nature of trajectory prediction.

Figure 6 shows the predicted trajectories of our method comparing with NMRF (Fang et al., 2025), and the ground-truth (GT) trajectories on NBA dataset. We see that PatchTraj enables to produce accurate results close to the real basketball players' movements in such a high density scene with diverse interactions.

These visual analyses complements our quantitative results, confirming PatchTraj's enhanced capability in capturing both individual motion characteristics and social navigation patterns.

## A.8 LIMITATIONS

On the NBA dataset, PatchTraj achieves the best ADE but does not outperform all baselines in FDE. This suggests that while our framework excels at capturing short-term dynamics through patch-level decomposition and deformable multi-scale fusion, it is less effective in modeling long-horizon intentions such as global destination choices or strategic positioning. Explicit intention modeling modules, such as goal-conditioned predictors or interaction-aware intention decoders, could potentially complement our approach. We regard this as a promising direction for future work, where PatchTraj's patch-level representation can serve as a fine-grained backbone while explicit intention modules provide stronger global constraints.

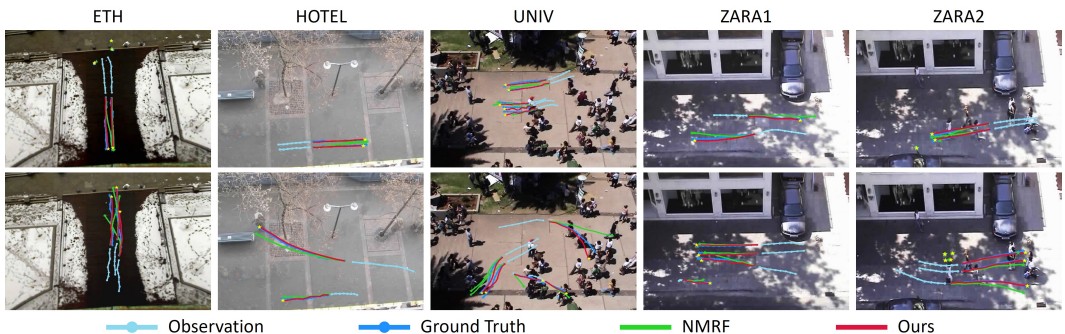

Figure 4: Visualization results on ETH-UCY dataset. Given past observations (sky blue), we compare the best predicted trajectory from 20 samples by NMRF (green) and our method (red). The ground truth future trajectory is shown in blue line. PatchTraj (Ours) predicts more robust trajectories than NMRF, with all predictions aligning closely with actual motion patterns.

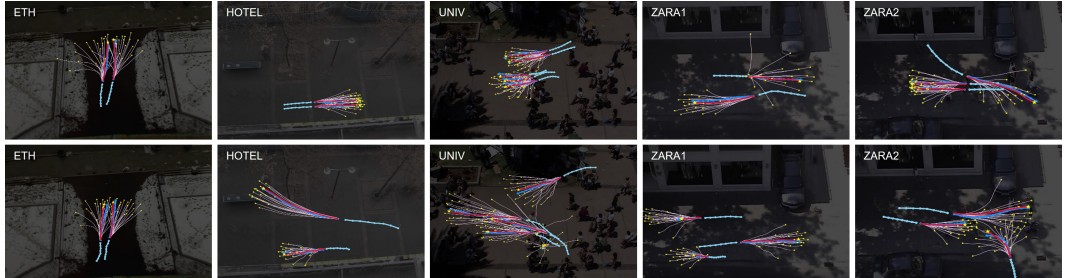

Figure 5: Visualization results of multi-modal prediction on ETH-UCY dataset. Given past observations (sky blue), we visualize the best of 20 predictions (red) produced by our method. The ground truth future trajectory is presented in blue line.

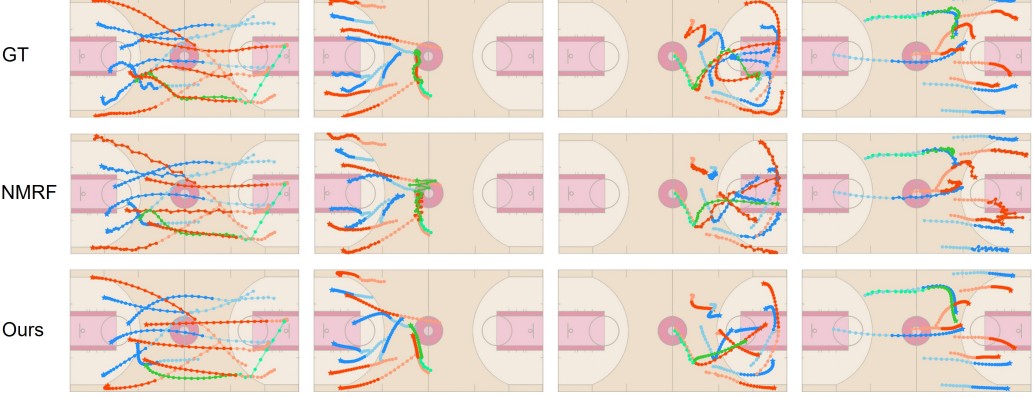

Figure 6: Visualization results on NBA dataset. Two team players are represented by blue and red color, basketball is represented by green color. Overall, light color indicates the past observations and dark color indicates the best predicted trajectories of 20 samples.

