# OpenReview forum: "PatchTraj: Unified Time-Frequency Representation Learning via Dynamic Patches for Trajectory Prediction"
_ICLR.cc/2026/Conference — ICLR 2026 Conference Withdrawn Submission_

### Official Review · Reviewer_TXzp · 2025-10-20

**Soundness:** 3
**Presentation:** 3
**Contribution:** 2
**Rating:** 4
**Confidence:** 4

**Summary:**

PatchTraj introduces a dynamic patch-based framework for pedestrian trajectory prediction that jointly models time and frequency domains. It adaptively segments trajectories into semantically coherent spatiotemporal patches, enabling hierarchical feature learning across multiple scales. The model employs expert-inspired multi-scale embeddings, deformable patch attention for cross-scale fusion, and cross-domain attention to integrate temporal and spectral cues.

**Strengths:**

- This study faithfully followed the experimental protocols of existing human trajectory prediction models. It utilized major benchmark datasets (JRDB, NBA, SDD, ETH-UCY) and adopted standard evaluation metrics (e.g., ADE, FDE), with results effectively visualized.

- Compared to prior methods, it achieved state-of-the-art performance and enhanced credibility by providing the implementation code in the supplementary materials. Additionally, extensive ablation studies on model architecture, loss functions, and the number of experts demonstrated the robustness and versatility of PatchTraj.

**Weaknesses:**

- While the paper compares PatchTraj with general human trajectory prediction methods, it lacks comparative experiments with conceptually or methodologically similar approaches. The authors mention TimesNet in line 145, yet no comparison is made with other models that also learn in the frequency domain (e.g., those based on FFT). Including such experiments would help validate the logical rationale for employing the Discrete Cosine Transform (DCT).

- Furthermore, the study provides insufficient in-depth comparisons with existing trajectory prediction methods. Although the reported results show generally superior performance, there is no detailed analysis comparing PatchTraj against point-based or grid-based methods from the Figure 1. Such experiments could reveal in which specific scenes or scenarios PatchTraj performs better, and why these advantages arise.

- The proposed approach appears somewhat naive and lacks strong technical novelty. The authors claim contributions through time–frequency hybridization, dynamic patch mechanisms, and multi-scale feature fusion, yet these techniques largely appear to be combinations of existing methods. Even if such integration is meaningful, the paper does not clearly explain 1) why these specific techniques were chosen over other similar approaches (and why others were not suitable), and 2) what modifications were made to adapt them to the human trajectory prediction domain (and what challenges arise when applying existing methods directly). Only the DPAttn module provides such clarification.

**Questions:**

- The authors mention the limitations of point-based and grid-based methods in the abstract, yet no clear empirical or conceptual evidence is provided. Please include supporting experiments or scenario-specific analyses to substantiate these claims.

- Are there any comparative experiments with prior point-based and grid-based approaches? Adding such comparisons would clarify the technical distinctions between PatchTraj and existing methods, thereby strengthening the paper’s contribution and justification.

- In line 45, the authors refer to “periodic patterns (e.g., gait cycles).” What specific periodic patterns does this work address? Are these patterns interpretable at a human-perceptible level (like gait cycles), or are they latent patterns beyond human interpretation?

- The paper employs the Discrete Cosine Transform (DCT) for time–frequency representation. What motivated this specific choice? Could other frequency transformation methods serve the same purpose, and if so, why was DCT preferred?

- While DCT is used to handle frequency information, there are no visual analyses demonstrating its effect. Including qualitative visualizations in the frequency domain would help readers verify that the model effectively learns and utilizes frequency features.

- The description of the MSPE module remains abstract. Please elaborate on its technical implementation, specifying how it differs from prior Mixture-of-Experts approaches and what these differences imply in the context of PatchTraj.

- According to Figure 2, the Time and Frequency branches share an identical structure. Are there visualizations showing what distinct representations each branch learns during training?

- In line 189, the term “contextual information” is ambiguous. Does it refer to the image scene at a given timestamp, or some other form of contextual data? Please clarify what this information specifically represents from an implementation perspective.

**Details Of Ethics Concerns:**

The dataset used in this study does not raise any ethical concerns.

---

### Official Review · Reviewer_yTNa · 2025-10-23

**Soundness:** 3
**Presentation:** 1
**Contribution:** 3
**Rating:** 4
**Confidence:** 4

**Summary:**

The paper proposes PatchTraj, and its core idea combines both the time domain and the frequency domain, and enhances the embedding so that the prediction can benefit from the fused knowledge. The good performance across multiple datasets validates the effectiveness of this framework. The key issue is the writing and presentation.

**Strengths:**

1. The model achieves state-of-the-art performance on four diverse datasets, outperforming recent baselines like NMRF (2025). It's interesting to see that both the ADE/FDE and the JADE/JFDE are reported for some dataset.
2. A detailed ablation study systematically evaluates the contribution of each major component of their proposed architecture, and the paper discusses the effect of K and joint loss variants.
3. The motivation to better unify local and global context and to explore the underutilized frequency domain is sound.

**Weaknesses:**

1. The presentation needs a lot of improvement.

E.g., in Figure 1, it's really difficult to understand the proposed Dynamic patching. What are s1, s2, and sM? What's the meaning of different rectangles?

In row 083, what is DPAttn? The author should at least show the full name of the module when using it for the first time.

Figure 2 is inconsistent; it is very difficult to understand how the two proposed branches are used in the right high-level architecture.

What is the dynamic patch in this figure? Overall, I think the author should clearly explain or define 'patch' in the very beginning.

2. Overclaimed contribution.

The authors claimed their time-frequency approach enables "noise-robust trajectory modeling". This is an empirical claim that is never tested. The paper lacks experiments about robustness against noise. For instance, adding synthetic noise to input trajectories to demonstrate that PatchTraj is more robust than time-domain-only baselines.

Also, the contribution about  'significantly outperforms previous state-of-the-art methods on four real-world datasets' is overstated, since the improvements of minFDE on NBA/SDD/ETH-UCY are not 'significant'.

**Questions:**

What is the inference speed of this model? Reporting the inference time or latency will make it more practical for applications.

---

### Official Review · Reviewer_mnwd · 2025-10-31

**Soundness:** 1
**Presentation:** 3
**Contribution:** 2
**Rating:** 2
**Confidence:** 5

**Summary:**

PatchTraj is a trajectory-prediction framework that represents motion jointly in time and frequency and replaces fixed windows with adaptive “patches”. It splits each input into a raw time-domain sequence and a DCT-based sequence, then a network groups the sequence into variable-length patches that better match motion segments. Each patch is embedded with a Mixture-of-Experts layer. Multi-scale features are aggregated, and the time and frequency branches exchange information through cross-modal attention. Then, a Transformer encoder–decoder predicts future timesteps. The authors claim that their method keeps fine local detail while modeling long-range dependencies and reduces noise sensitivity that affects pure time-domain methods.

**Strengths:**

- The paper is clearly written. The problem statement, motivation and solution are explained well.

- The core idea is novel. The experimental setup is described clearly and organized coherently.

- Hyperparameter choices and implementation details are provided in the appendix, which, together with the released code, supports reproducibility.

**Weaknesses:**

- The authors claim that incorporating frequency data improves long-term dependency modeling, yet the FDE gains from adding the F-branch (Table 2) are limited compared to this claim.

- The paper emphasizes noise robustness, but this is not evaluated.

- More ablations would be beneficial: variants that (i) assign patch sizes randomly and (ii) restrict to a small set of reasonable patch sizes.

**Questions:**

- How is the number of retained coefficients chosen, and is performance sensitive to this choice?

- Does padding with the last observed position ever bias the frequency branch toward “stopped” motion?

- Do different patch scales actually activate different experts in practice, or does the gate collapse to a few experts?

---

### Official Review · Reviewer_nDtp · 2025-11-01

**Soundness:** 3
**Presentation:** 2
**Contribution:** 2
**Rating:** 4
**Confidence:** 3

**Summary:**

This paper proposes PatchTraj, a pedestrian trajectory forecasting framework that jointly models time and frequency domains using dynamic patching. It splits each trajectory into multi-scale patches, and fuses them after encoding. Across multiple datasets, PatchTraj shows state-of-the-art accuracy. Ablation study shows each module’s contribution.

**Strengths:**

1. The idea of fusing time and frequency is well motivated and intuitive.
2. The architecture is clearly presented and well motivated.
3. The results across multiple datasets are strong and state of the art.

**Weaknesses:**

1. The method put together several known ideas into one large system, laying out this multi-module pipeline. It’s more like a comprehensive engineering package than an innovative idea.
2. The ablation study shows additive gains as each module is turned on. It doesn’t show which subset has better accuracy per cost tradeoff. If we add more modules, would the performance be even better? In this sense, we’d like to understand how much additional cost, e.g. inference latency, the pipeline incurs relative to the baseline.
3. The paper claims it learns to group trajectory points into semantically meaningful patches based on motion dynamics, but the method section defines non-overlapping fixed patch sizes chosen from a set S. There is no learning involved.

**Questions:**

There are many typos or grammar mistakes. To list a few:
1. Line 062 to Line 070 is not a complete sentence.
2. Line 081: “where learns” → “which learns”.
3. Line 905 to Line 907: “We also conduct … our method.” are two sentences. Also “investigate” → “investigates”.

---

### Note · Authors · 2025-11-22

I have read and agree with the venue's withdrawal policy on behalf of myself and my co-authors.